# Detection of CWD in cervids by RT-QuIC assay of third eyelids

**Sarah K. Cooper[1], Clare E. Hoover[1¤], Davin M. Henderson[1], Nicholas J. Haley[2], Candace K. Mathiason[1], Edward A. Hoover[1]** *

**1** Prion Research Center, Department of Microbiology, Immunology, and Pathology, College of Veterinary Medicine and Biomedical Sciences, Colorado State University, Fort Collins, Colorado, United States of America, **2** Department of Microbiology and Immunology, Midwestern University, Glendale, Arizona, United States of America

¤ Current address: AstraZeneca, Waltham, Massachusetts, United States of America
* edward.hoover@colostate.edu

**Data Availability Statement:** All relevant data are within the manuscript.

**Funding:** This work received support from the Foundation for the National Institutes of Health, R01-NS061902, to Dr. Edward A. Hoover; the

## Abstract

The diagnosis of chronic wasting disease (CWD) relies on demonstration of the disease-associated misfolded CWD prion protein ($PrP^{CWD}$) in brain or retropharyngeal lymph node tissue by immunodetection methods, e.g. ELISA and immunohistochemistry (IHC). The success of these methods relies on a quality sample of tissues, which requires both anatomical knowledge and considerable dissection to collect. As the prevalence of CWD continues to increase globally, the development of fast and cost-effective methods to detect the disease is vital to facilitate CWD detection and surveillance. To address these issues, we have evaluated third eyelids from CWD-infected deer and elk using real-time quaking induced conversion (RT-QuIC). We identified prion seeding activity in third eyelids in 24 of 25 (96%) CWD-infected white-tailed deer (*Odocoileus virginianus*). We detected RT-QuIC positivity in the third eyelid as early as 1 month after experimental CWD exposure. In addition, we identified prion seeding activity in third eyelids of 18 of 25 (72%) naturally exposed asymptomatic CWD-positive rocky mountain elk (*Cervus canadensis nelson*). We compared CWD detection by RT-QuIC and IHC in third eyelid, retropharyngeal lymph node, and brain in 10 deer in early symptomatic stage of disease. IHC detected $PrP^{CWD}$ deposition in third eyelid lymphoid follicles in 5 of 10 deer (50%) whereas third eyelids of all 10 animals were positive by RT-QuIC. This difference reflected in part a lower requirement for lymphoid follicle presence for seeding activity detection by RT-QuIC. In conclusion, RT-QuIC analysis of the third eyelid, an easily accessed tissue, has potential to advance CWD detection and testing compliance.

## Introduction

Chronic Wasting Disease (CWD) is a fatal contagious prion disease affecting cervid species (deer, elk, and moose) that is characterized by neurodegeneration, emaciation, and abnormal behaviors [1–5]. CWD, first identified in Colorado, now is found in North America, South

Foundation for the National Institutes of Health, P01-AI-077774, to Dr. Edward A. Hoover; and the Foundation for the National Institutes of Health, R01-AI112956, to Dr. Candace K. Mathiason. The funders had no role in study design, data collection and analysis, decision to publish, or preparation of the manuscript.

**Competing interests:** The authors have declared that no competing interests exist.

Korea, and Scandinavia [3–5]. Natural infection and transmission likely occurs through oral and nasal mucosal contact with infectious prions [1, 6–11]. Disease begins when the infectious prion induces continuous misfolding of the normal cellular protein (PrP$^C$) into a disease-associated, protease-resistant form (PrP$^{RES}$) which aggregates into amyloid fibrils [12–14]. In several prion diseases, including scrapie in sheep and CWD in cervids, prions accumulate first in the systemic lymphoid tissues before entering the central nervous system [7, 15–17]. Polymorphisms in the *PRNP* gene sequence (most notably G96S in deer and M132L in elk) are known to prolong the disease course [3, 18]. The more common *PRNP* G96G and M132M genotypes are the most common polymorphisms and are associated with higher frequencies of CWD infection in deer and elk [19, 20].

The standard diagnosis of CWD is identification of CWD protease resistant prion protein (PrP$^{CWD}$) by immunodetection methods such as ELISA and IHC [21]. More recently developed real-time seeding and amyloid amplification methods have increased CWD detection sensitivity [2, 15, 22]. In real-time quaking induced conversion (RT-QuIC) a prion seed converts recombinant normal prion protein, PrP$^C$, into amyloid fibrils, an event detectable by the binding of thioflavin T (ThT). Alternating cycles of incubation and shaking are used to facilitate fibril fragmentation and re-seeding, thus amplifying minute amounts of prion seed to a detectable level [23–25]. Previous studies have validated RT-QuIC for the identification of PrP$^{CWD}$ in brain, lymph nodes, and other tissues, as well as in secretions and excretions [11, 15].

The increasing prevalence of CWD globally makes critical the development of fast, cost-effective methods to detect the disease in deer and assist with disease management. The third eyelid is a nictitating membrane found in many animal species located between the globe of the eye and the lower eyelid, thereby easily accessible without special anatomical training [26]. In ruminants, including cervids, the membrane contains lymphoid tissue organized into the lymphoid follicles with germinal centers where prion protein can accumulate at early stages of disease [26, 27]. Here we have explored the potential of the third eyelid for rapid detection of CWD infection, based on the work of O'Rourke et al [28, 29] for detection of scrapie in sheep. We use both RT-QuIC and IHC examination of third eyelids to detect CWD infection in symptomatic and pre-symptomatic white-tailed deer and rocky mountain elk to demonstrate the utility of this accessible tissue for rapid diagnosis of CWD in cervids.

## Results

### RT-QuIC analysis of third eyelids from symptomatic deer

To evaluate whether RT-QuIC could detect CWD in third eyelid tissue, we examined third eyelids collected at necropsy from n = 25 white-tailed deer (WTD) experimentally exposed to CWD-positive saliva or brain homogenate, usually by the oral, or in one study, the aerosol route [27]. Dose protocols included oral inoculation of either: (a) 300ng, 0.001g, 0.01g, or 1.0g of CWD-positive brain homogenate; or (b) 30 mL of CWD-positive saliva containing 300ng brain equivalent seeding activity in RT-QuIC; or (c) aerosolization of 0.1g of CWD-positive brain homogenate [30, 31]. CWD infection was confirmed in all of these animals by IHC detection of PrP$^{CWD}$ in the obex region of the brain and the retropharyngeal lymph nodes (RPLN). Third eyelid homogenates from 20 of 21 deer (95%) containing the *PRNP* codon 96GG genotype displayed significant amyloid seeding activity by RT-QuIC (****p < 0.0001, two-tailed Mann-Whitney test vs. negative control eyelids) (Fig 1A–1D). The same 21 deer also were positive for RT-QuIC seeding activity in obex and RPLN. Third eyelids from the 4 deer of 96GS genotype also demonstrated significant amyloid seeding activity by RT-QuIC (****p < 0.0001, two-tailed Mann-Whitney test) (Fig 1A) and were likewise positive in

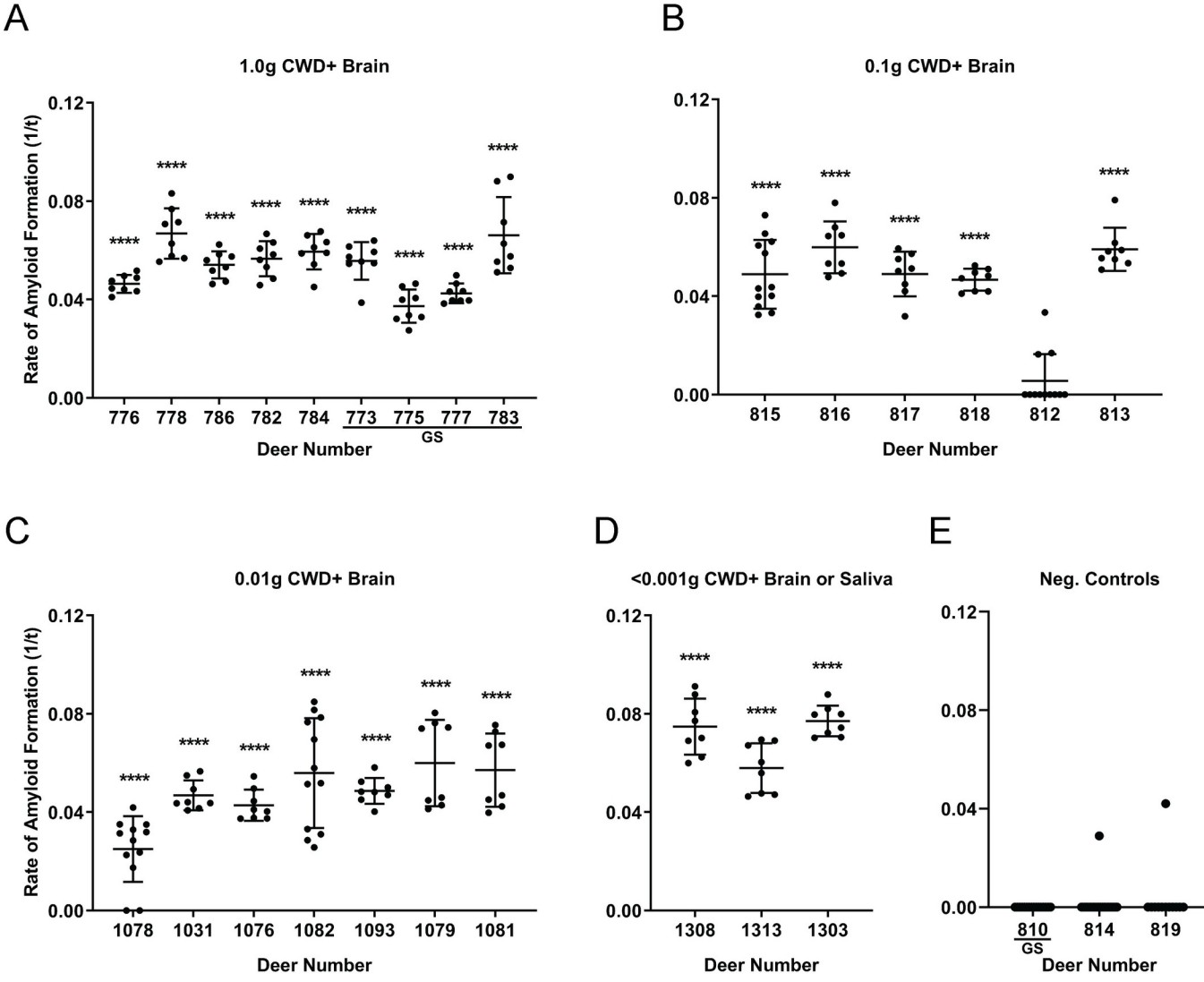

**Fig 1. Detection of PrP^CWD in third eyelids of symptomatic deer by RT-QuIC.** (A) RT-QuIC analysis of third eyelid samples collected from 96GG and 96GS WTD inoculated with 1.0g of CWD-positive deer brain by oral administration (*per os*). Statistically significant amyloid seeding activity was detected in the third eyelids of all the deer in this cohort (****p < 0.0001, two-tailed Mann-Whitney test) when compared with respective negative control tissues. (B) RT-QuIC analysis of third eyelid samples collected from 96GG WTD inoculated with 0.1g of CWD-positive deer brain via aerosolization. 5 of 6 third eyelids displayed significant amyloid seeding activity (****p < 0.0001, two-tailed Mann-Whitney test) when compared with respective negative control tissues. (C) RT-QuIC analysis of third eyelid samples collected from 96GG WTD inoculated with 0.01g of CWD-positive deer brain *per os*. 3 of 3 third eyelids displayed significant amyloid seeding activity (****p < 0.0001, two-tailed Mann-Whitney test) when compared with respective negative control tissues. (D) RT-QuIC analysis of third eyelid samples collected from 96GG WTD inoculated with 300ng-0.001g of brain or saliva *per os* respectively. (E) RT-QuIC analysis of third eyelid sample controls collected from one 96GS and two 96GG WTD inoculated with 0.1g of CWD-negative deer brain via aerosolization for negative controls. Each third eyelid sample is represented by the mean and standard deviation from at least eight replicates.

systemic tissues. False positive wells in negative control third eyelids were well below a level of significance (2 false positive replicates of 44 total replicates (4.5%) (Fig 1E). These results demonstrated that the third eyelid can be used in RT-QuIC assay to consistently detect PrP^CWD amyloid seeding activity from a variety of CWD-infected, symptomatic WTD with little false positivity. Additionally, we found that detection of amyloid seeding activity in the third eyelid is not confined to 96GG genotype, demonstrated by the seeding activity found in all four 96GS deer.

## IHC analysis of third eyelids from 96GG terminal deer

To further explore the efficacy of the third eyelid in CWD detection, we examined paraformaldehyde-fixed tissues from n = 10 96GG deer of the above 21 animals for PrP$^{CWD}$ IHC immunoreactivity in the obex, RPLN, and the third eyelid (Fig 2). Despite RT-QuIC identifying PrP$^{CWD}$ seeding activity in third eyelids of 20 of 21 (95%), only 5 of the 10 (50%) third eyelid samples demonstrated PrP$^{CWD}$ immunoreactivity in germinal centers of the limited number of lymphoid follicles present (Fig 1 and Table 1). Clear PrP$^{CWD}$ staining was seen in follicles of the third eyelid and RPLN and larger aggregate plaques were present in the obex (Fig 2). These results demonstrated that RT-QuIC detection of seeding activity in third eyelids from CWD-infected, symptomatic WTD correlated with IHC positivity in retropharyngeal lymph node and brain obex samples. Additionally, IHC showed less consistent detection in the third eyelid when compared with RT-QuIC, in part due to insufficient presence of lymphoid follicles in the tissue.

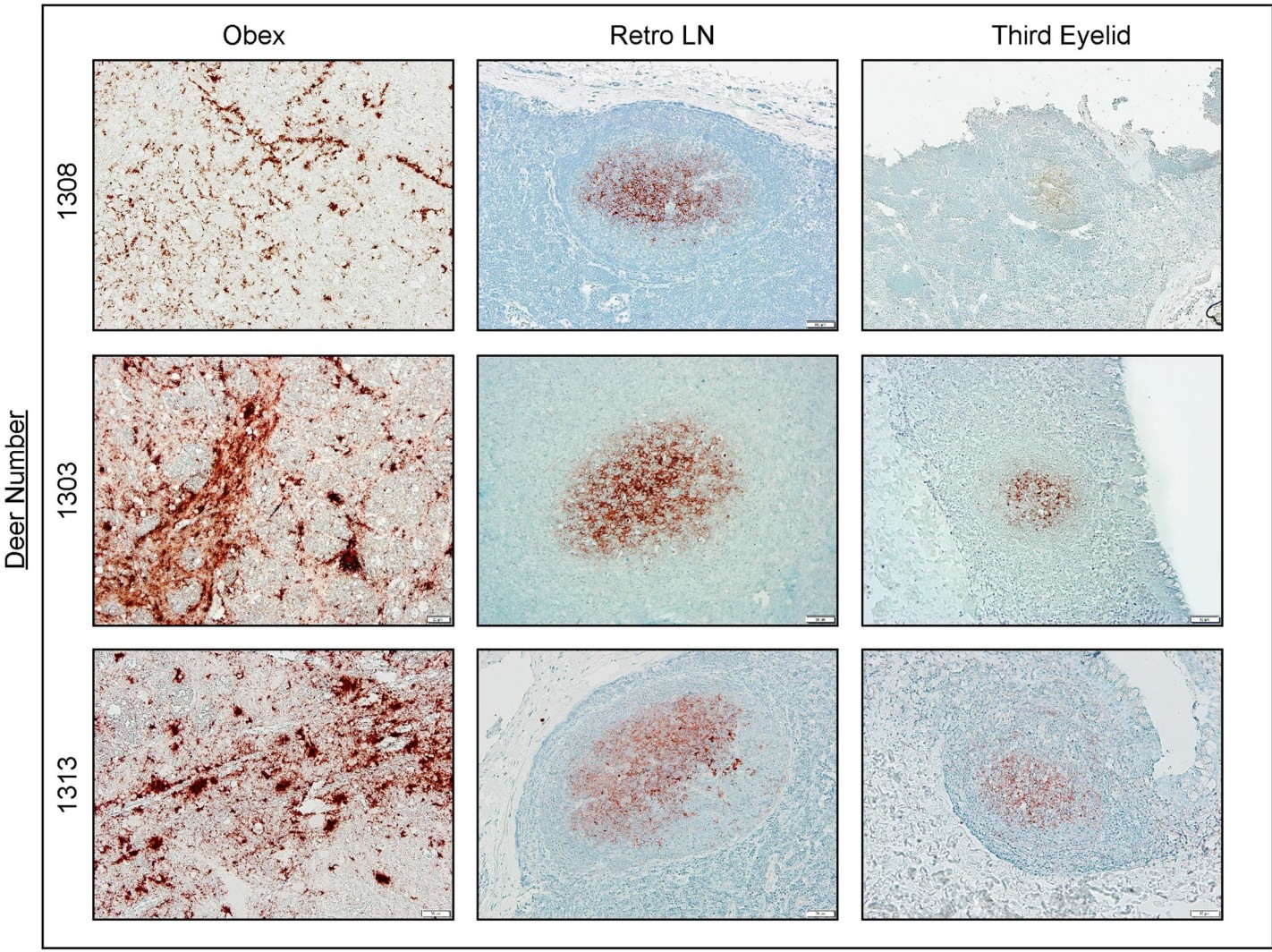

**Fig 2. Detection of PrP$^{CWD}$ in terminal 96GG deer by IHC.** PrP$^{CWD}$ immunoreactivity was present in plaques in the obex and in germinal centers of lymphoid follicles of the retropharyngeal lymph node and third eyelid. Displayed are 3 positive deer of the 10 deer assayed by both IHC and RT-QuIC. IHC images are 200X magnification; scale bar = 50 μm. Abbreviations: Retro LN, retropharyngeal lymph node; IHC, immunohistochemistry.

**Table 1. Detection of PrP$^{CWD}$ in pre-symptomatic deer by RT-QuIC and IHC.**

| Deer Number | RT-QuIC | | | IHC | | |
|---|---|---|---|---|---|---|
| | Obex | RPLN | Third Eyelid | Obex | RPLN | Third Eyelid |
| 1076 | + | + | + | + | + | + |
| 1082 | + | + | + | + | + | + |
| 1093 | + | + | + | + | + | - |
| 1078 | + | + | + | + | + | - |
| 1079 | + | + | + | + | + | - |
| 1031 | + | + | + | + | + | - |
| 1081 | + | + | + | + | + | - |
| 1303 | + | + | + | + | + | + |
| 1308 | + | + | + | + | + | + |
| 1313 | + | + | + | + | + | + |

RT-QuIC detection of PrP$^{CWD}$ seeding activity was comparable to IHC when evaluating obex and retropharyngeal lymph node samples. However, RT-QuIC analysis detected positivity in more third eyelid samples, 10 of 10 (100%), compared with 5 of 10 (50%) by IHC. Data are presented as animals PrP$^{CWD}$ positive or negative by tissue type and detection method. Samples were deemed RT-QuIC positive if they achieved statistical significance ($p < 0.0001$, two-tailed Mann-Whitney test) when compared with respective negative control tissues. Abbreviations: RPLN, retropharyngeal lymph node.

## RT-QuIC analysis of third eyelids from pre-symptomatic deer

In order to determine whether third eyelids accumulated PrP$^{CWD}$ early in the disease course, we evaluated samples from two groups of asymptomatic deer. The first group of pre-symptomatic deer were collected 1 to 4 months after oral CWD inoculation (0.5 g of CWD-positive brain) [15]. In these deer, RT-QuIC seeding activity and IHC immunoreactivity was detected in upper alimentary tract lymphoid tissues as early as 1-month post-inoculation but was not detected in the obex region of the brain (Table 2). In one 96GG deer at 1-month post-inoculation (MPI) and 4 of 8 96GG additional deer at 2, 3, and 4 MPI, third eyelids contained significant amyloid seeding activity (*$p < 0.05$, two-tailed Mann-Whitney test) (Fig 3 and Table 2). In n = 3 96GS deer, PrP$^{CWD}$ was not significantly detected in third eyelids at these same early time points. This latter disparity likely reflects the slower disease onset and course in this genotype as seeding activity was also not detected in retropharyngeal lymph nodes of these animals (Fig 3 and Table 2) [15]. These results demonstrated that in addition to detecting significant third eyelid amyloid seeding activity at early time points in 96GG deer, detection was also possible in the third eyelid of an asymptomatic 96SS genotype deer.

## RT-QuIC and IHC analysis of third eyelids from pre-symptomatic deer

We examined a second cohort of 5 deer necropsied between 18 and 27 months after low-dose oral CWD inoculation (300 ng or 0.001g CWD-positive brain homogenate; or 30 mL of CWD-positive saliva containing 300ng brain equivalent seeding activity in RT-QuIC), but had not displayed clinical signs of CWD. Significant prion seeding activity was detected in retropharyngeal lymph node and obex of all 5 deer and in third eyelid of 4 of 5 (80%) deer, including one 96GS deer (****$p < 0.0001$, two-tailed Mann-Whitney test) (Fig 4). Retropharyngeal lymph nodes from all 5 deer and obex samples from 4 of the 5 deer (80%) demonstrated PrP$^{CWD}$ immunoreactivity by IHC (Fig 4). PrP$^{CWD}$ immunoreactivity was detected in third eyelid lymphoid follicles in 3 of 5 deer (60%) including 1 96GS deer by IHC (Fig 4). These results demonstrated slightly higher detection in the third eyelid by RT-QuIC compared with IHC in pre-symptomatic deer exposed to low doses of CWD-positive material.

**Table 2. Detection of PrP^CWD in deer by RT-QuIC.**

| Deer Number | Tissue | | | MPI | Clinical Status | Genotype |
|---|---|---|---|---|---|---|
| | RPLN | Obex | Third Eyelid | | | |
| 1174 | - | - | - | 1 | PS | GG |
| 1202 | + | - | - | 1 | PS | GG |
| 1219 | + | - | + | 1 | PS | GG |
| 1167 | - | - | - | 1 | PS | GS |
| 1157 | + | - | - | 2 | PS | GG |
| 1148 | + | - | - | 2 | PS | GG |
| 1218 | + | - | + | 2 | PS | GG |
| 1204 | - | - | - | 2 | PS | GS |
| 1207 | + | - | + | 3 | PS | GG |
| 1203 | + | - | + | 3 | PS | GG |
| 1215 | + | - | + | 4 | PS | GG |
| 1201 | + | - | - | 4 | PS | GG |
| 1171 | + | - | - | 4 | PS | GG |
| 1156 | - | - | - | 4 | PS | GS |
| 1205 | + | - | + | 16 | PS | SS |
| 776 | + | + | + | 27 | S | GG |
| 778 | + | + | + | 22 | S | GG |
| 786 | + | + | + | 16 | S | GG |
| 782 | + | + | + | 22 | S | GG |
| 784 | + | + | + | 18 | S | GG |
| 773 | + | + | + | 30 | S | GS |
| 775 | + | + | + | 31 | S | GS |
| 777 | + | + | + | 31.5 | S | GS |
| 783 | + | + | + | 16 | S | GS |
| 815 | + | + | + | 23 | S | GG |
| 816 | + | + | + | 25 | S | GG |
| 817 | + | + | + | 19 | S | GG |
| 818 | + | + | + | 16.5 | S | GG |
| 812 | + | + | - | 26 | S | GG |
| 813 | + | + | + | 23 | S | GG |
| 1078 | + | + | + | 24.5 | S | GG |
| 1031 | + | + | + | 32 | S | GG |
| 1076 | + | + | + | 22 | S | GG |
| 1082 | + | + | + | 24 | S | GG |
| 1093 | + | + | + | 23 | S | GG |
| 1081 | + | + | + | 20 | S | GG |
| 1308 | + | + | + | 18 | S | GG |
| 1313 | + | + | + | 25 | S | GG |
| 1303 | + | + | + | 22 | S | GG |
| 1305 | + | + | - | 28 | PS | GG |
| 1309 | + | + | + | 28 | PS | GG |
| 1316 | + | + | + | 23 | PS | GG |
| 1307 | + | + | + | 28 | PS | GG |

*(Continued)*

**Table 2.** (Continued)

| Deer Number | Tissue | | | MPI | Clinical Status | Genotype |
|---|---|---|---|---|---|---|
| | RPLN | Obex | Third Eyelid | | | |
| 1310 | + | + | + | 28 | PS | GS |

Summary of PrP$^{CWD}$ detection in WTD by RT-QuIC. Data are presented as animals PrP$^{CWD}$ positive or negative by tissue type with respective months post inoculation, clinical status, and PRNP codon 96 genotype: homozygous G96G (GG) and heterozygous G96S (GS), homozygous S96S (SS). Samples were deemed RT-QuIC positive if statistical significance was demonstrated ($p < 0.05$, two-tailed Mann-Whitney test) when compared with respective negative control tissues. Abbreviations: RPLN, retropharyngeal lymph node; MPI, months post inoculation; PS, pre-symptomatic; S, symptomatic.

### RT-QuIC analysis of third eyelids from asymptomatic elk

To investigate whether PrP$^{CWD}$ accumulates in third eyelids of other cervid species, we examined samples from asymptomatic, naturally exposed elk. The elk were culled due to positive or suspect results on RAMALT tissue biopsies by IHC or RT-QuIC [32]. Statistically significant prion seeding activity was detected in 18 of 25 (72%) third eyelids (*$p < 0.05$, two-tailed Mann-Whitney test compared with controls) (Fig 5). We found significant seeding activity in 12 of 16 (75%) of 132MM genotype elk, 5 of 8 (63%) of 132ML genotype, and 1 of 1 (100%) of 132LL genotype. 22 retropharyngeal lymph nodes and 23 obex samples were tested by IHC. PrP$^{CWD}$ immunoreactivity was observed in all 22 retropharyngeal lymph nodes and in 20 of the 23 obex samples (Fig 5). Significant seeding activity was detected by RT-QuIC in 15 of 19 (79%) third eyelid samples from asymptomatic elk that were IHC positive in both obex and RPLN. These results demonstrated that PrP$^{CWD}$ can be detected in third eyelids of elk with differing codon 132 genotypes, even at asymptomatic stages of disease.

## Discussion

The work of O'Rourke and colleagues [5] demonstrated that scrapie prions can accumulate in third eyelid lymphoid germinal centers and constitute a USDA-approved method for scrapie

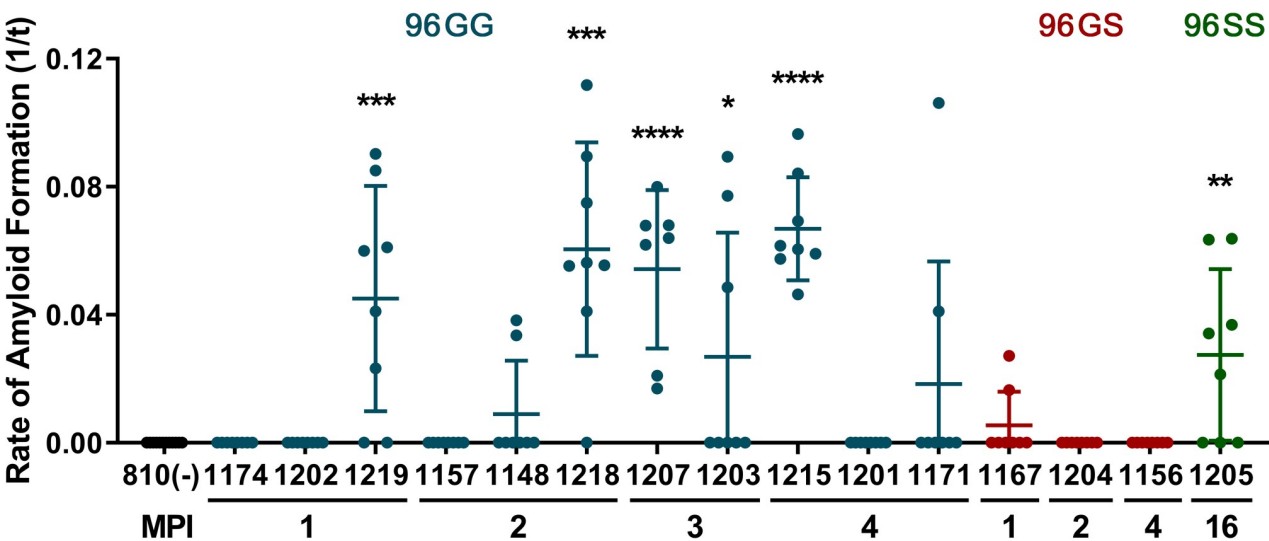

**Fig 3. Detection of PrP$^{CWD}$ in third eyelids of pre-symptomatic deer by RT-QuIC.** CWD prions were detected by RT-QuIC in the third eyelid of a 96GG deer as early as 1 month post-inoculation. RT-QuIC analysis of 96GG negative-control (black), 96GG (blue), 96GS (red), and 96SS (green) third eyelids collected in chronological order of months post-inoculation (MPI). 6 samples displayed statistically significant RT-QuIC amyloid seeding: 1219, 1218, 1207, 1203, 1215, and 1205 (****$p < 0.0001$, ***$p < 0.001$, **$p < 0.01$, *$p < 0.05$, two-tailed Mann-Whitney test) when compared with respective negative control tissues. Each third eyelid sample is represented by the mean and standard deviation from at least eight replicate wells.

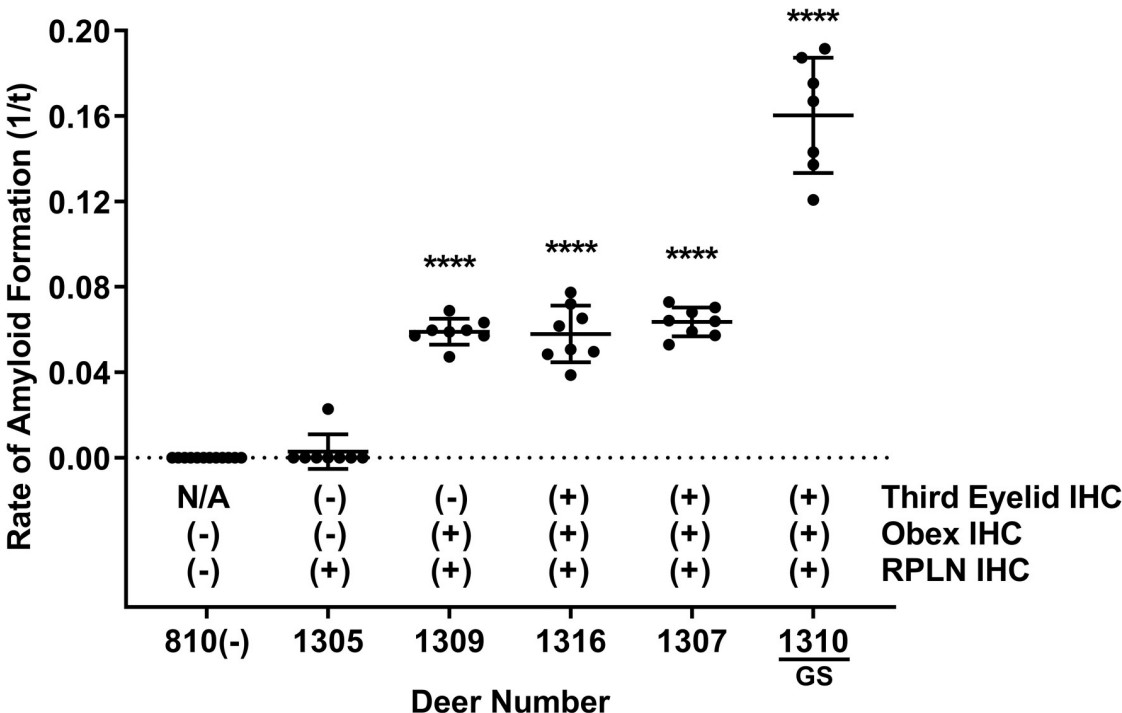

**Fig 4. Detection of PrP$^{CWD}$ in pre-symptomatic deer by RT-QuIC and IHC.** RT-QuIC analysis of third eyelid samples from four 96GG deer and one 96GS deer demonstrated significant amyloid seeding activity in 4 of 5 animals (80%) (****p < 0.0001, two-tailed Mann-Whitney test) when compared with respective negative control tissues, while only 3 of 5 (60%) were positive by IHC. RT-QuIC detection of PrP$^{CWD}$ was comparable with IHC in retropharyngeal lymph node and obex samples tested. Data are presented as PrP$^{CWD}$ positive or negative by tissue type and detection method. Each third eyelid sample is represented by the mean and standard deviation from at least eight replicates. Abbreviations: RPLN, retropharyngeal lymph node, IHC; immunohistochemistry.

diagnosis in sheep [28, 29]. We show here that the seeding of the third eyelid in deer may be a basis for detection of CWD as well. RT-QuIC detected CWD seeding activity in third eyelids as early as one month post inoculation and in most (28/29) deer and elk that were IHC positive in both obex and RPLN, even if they were not displaying clinical signs. In the present study, detection of CWD by IHC on third eyelids of deer was less sensitive (8/15) compared with detection of scrapie in sheep wherein 41/42 samples were positive [28]. There are several potential reasons for the detection discrepancy between these two methods in deer. One is that RT-QuIC employs whole homogenized samples, which increases the possibility that lymphoid content is included in the sample. Another is that lymphoid follicles are usually required for unequivocal IHC detection, which was compromised when follicle number was low. Finally, previous studies have demonstrated that RT-QuIC can enhance CWD detection in tissues or excreta samples in which prion concentrations are quite low [15, 22]. Thereby, the amplification provided by RT-QuIC may account for increased detection sensitivity in eyelids as well [22].

We observed higher CWD prion detection in deer eyelid samples compared with elk samples. One potential reason for this is that the elk samples used in this study were collected from asymptomatic elk in a CWD-infected herd involved in a herd management program and thereby a variety of early infection states may be been represented. By contrast, the majority of the deer used for this study were in or approaching a symptomatic disease state. In addition, previous investigations into CWD pathogenesis found greater PrP$^{CWD}$ accumulation in

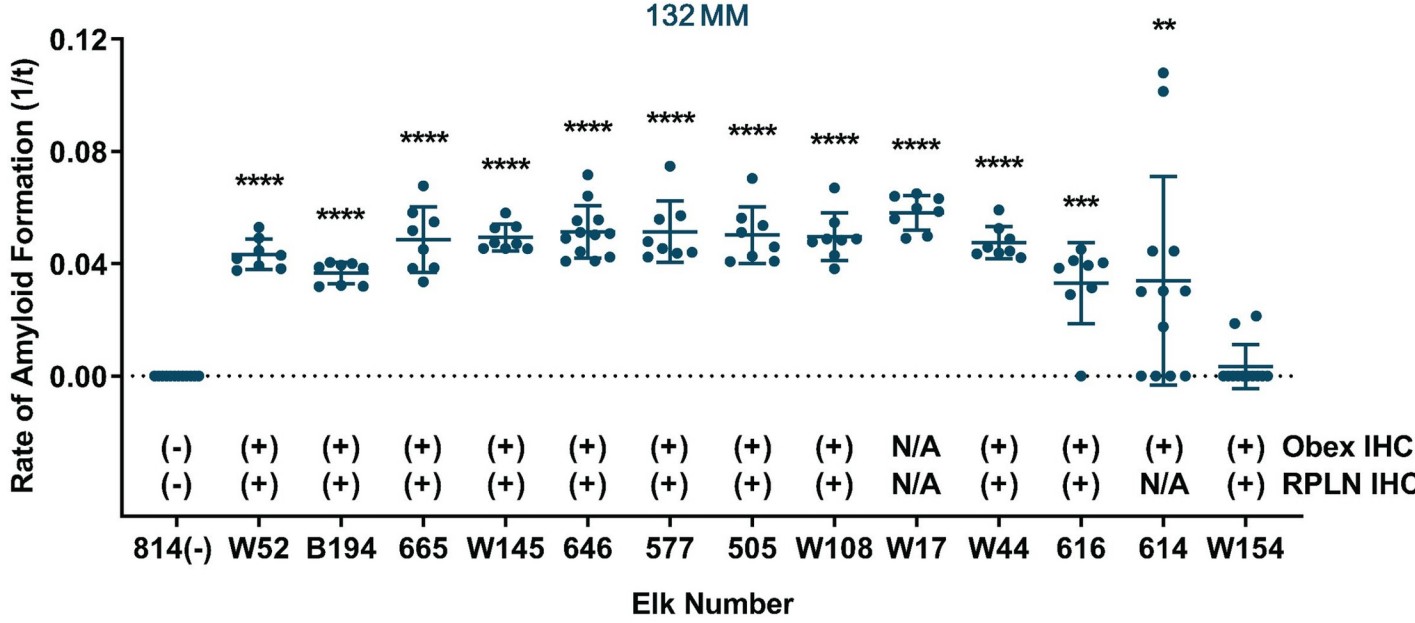

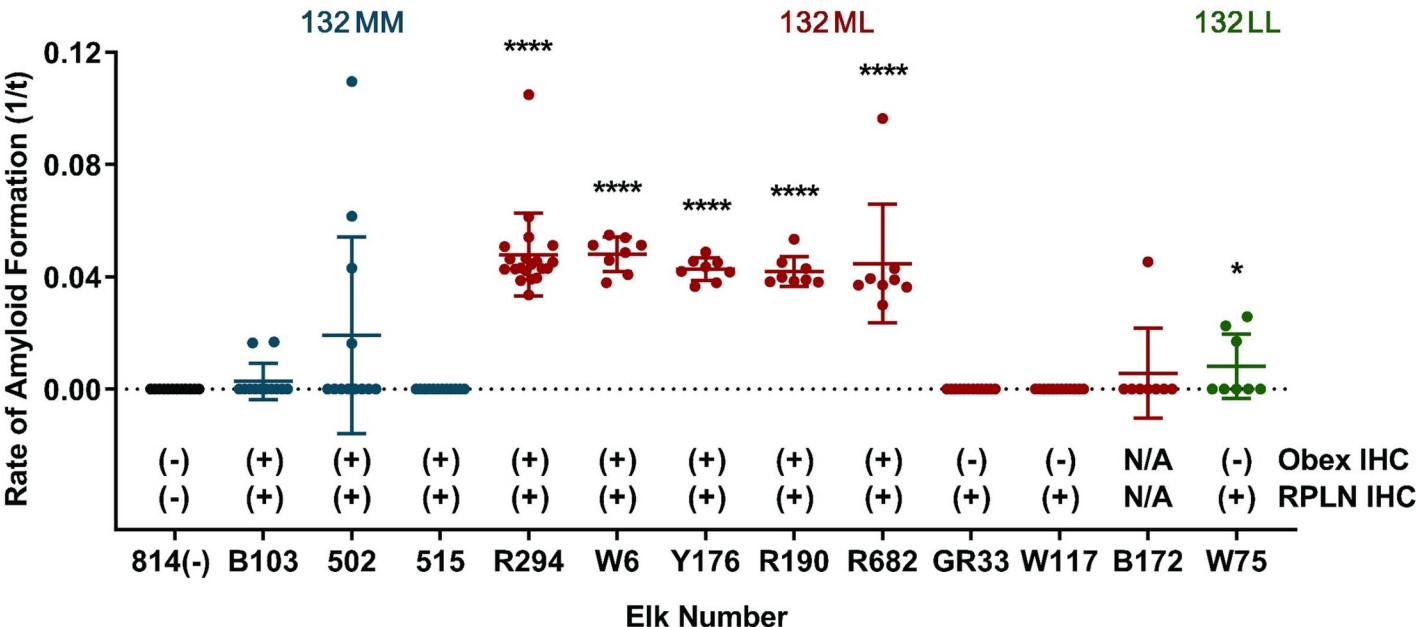

**Fig 5. Detection of PrP^CWD in asymptomatic elk by RT-QuIC and IHC.** RT-QuIC detection of PrP$^{CWD}$ in asymptomatic elk demonstrated statistically significant detection in third eyelids of 18 of 25 (72%) animals (****p < 0.0001, ***p < 0.001, **p < 0.01, *p < 0.05, two-tailed Mann-Whitney test compared with negative controls). PrP$^{CWD}$ immunoreactivity was detected in 22 of 22 (100%) retropharyngeal lymph node samples and 20 of 23 (87%) obex samples of by IHC. Data are presented as PrP$^{CWD}$ positive or negative by genotype: 132MM (blue), 132ML (red), 132LL (green), tissue type and test method. Each third eyelid sample is represented by the mean and standard deviation from at least eight replicates. Abbreviations: RPLN, retropharyngeal lymph node; IHC, immunohistochemistry.

lymphoid tissues of infected deer than do elk [33]. Deer also display a greater distribution of PrP$^{CWD}$ in lymphoid tissues compared with elk [34]. Such species differences in prion

lymphoid tissue accumulation may contribute to the variation in third eyelid detection and highlight the advantage of more sensitive testing methods in herd management programs.

In conclusion, we demonstrate that RT-QuIC performed on third eyelid tissue can be used to detect CWD in deer and elk, including those in pre-symptomatic stages of infection. As third eyelid is an easily accessible tissue, it has potential to aid in surveillance and screening programs.

## Materials and methods

### Deer care and inoculum

Hand-raised, indoor-adapted, white-tailed deer (*Odocoileus virginianus*) were sourced from the Warnell School of Forestry and housed in the CSU CWD Research Facility in biosecure indoor suites in strict accordance with Colorado State University approved Institution Animal Care and Use Committee protocols. The Colorado State University Institution Animal Care and Use has specifically approved this study. The PRNP genotype of each deer was determined to be homozygous G96G (96GG), heterozygous G96S (96GS), or homozygous S96S (96SS) at codon 96 as previously described [35]. Inoculation and maintenance protocols were followed as previously described to ensure proper dosing and avoid cross contamination between suites [36].

Anesthetized deer were inoculated with CWD-positive material orally by slow syringe installation with head position upright to minimize potential for aspiration and simulate natural oral exposure. Animal numbers and dosage were as follows:

i.  Deer #773, 775, 776, 777, 778, 782, 783, 784, 786 were inoculated with 1.0g of CWD-positive deer brain *per os* and were sacrificed between 16 and 32 months post inoculation as previously described [31].

ii.  Deer #812, 813, 815, 816, 817, 818 were inoculated with 0.1g of CWD-positive deer brain via aerosolization and were sacrificed in symptomatic stages of disease between 17 and 26 months post inoculation. 810, 814, 819 were inoculated with 0.1g of CWD-negative deer brain via aerosolization and were housed in different suites as the CWD-positive inoculated deer. These deer were sacrificed between 19 and 26 months post inoculation as previously described [30].

iii.  Deer #1093, 1076, 1083, 1082, 1078, 1079, 1031, 1081 were inoculated with 0.01g of CWD-positive deer brain *per os*. Deer were monitored and sacrificed between 20 and 32 months post inoculation when animals had symptomatic stages of disease.

iv.  Deer #1303, 1316, and 1307 were inoculated *per os* with a total of 300ng of CWD-positive deer brain in three separate weekly doses and were sacrificed at 22, 23, and 28 months post inoculation respectively. Deer #1313 and 1309 were inoculated *per os* with a total of 30 mL of CWD-positive deer saliva in three separate weekly doses and were sacrificed at 25 and 28 months post inoculation respectively. Deer #1308, 1310, and 1305 were inoculated with 0.001g *per os* of CWD-positive deer brain and were sacrificed at 18, 28 and 28 months post inoculation respectively.

v.  Deer #1167, 1174, 1157, 1148, 1204, 1207, 1201, 1215, 1156, 1171, 1205 were inoculated with 0.5g of CWD-positive deer brain *per os* and sacrificed in various stages of prepre-symptomatic disease as previously described [15].

## Deer tissue collection and processing

Each tissue was collected at necropsy with separate prion-free instruments and were divided into two halves; one-half was stored at -80 ˚C and the other half was fixed in periodate-lysine-paraformaldehyde (PLP) before being transferred to 1X phosphate-buffered saline (PBS) or 70% ethanol for long-term storage. Following fixation, tissues were trimmed and placed in histology cassettes, for processing into paraffin-embedded blocks. Third eyelids were trimmed longitudinally from the apical part of the tissue in proximity to where lymphoid tissue is located. The dissected third eyelid tissues were homogenized at 10% (wt/vol) in 1X PBS using a Bead Ruptor 24 (Omni International). The homogenates were stored at -80 ˚C until further use.

## Elk tissue collection and processing

Asymptomatic elk from a privately owned Colorado herd containing CWD-positive animals were euthanized and culled as part of a herd management program after rectal biopsies tested positive by RT-QuIC. Terminal elk IHC (brain and RPLN) testing was required and performed by the USDA in Ames Iowa. Third eyelid tissue samples were collected from elk with single use sterile instruments [29]. The tissues were then processed as described above for deer eyelids.

## RT-QuIC assay

**RT-QuIC substrate protein purification.** RT-QuIC assays were executed with 0.1mg/ml recombinant truncated Syrian hamster PrP$^C$ (SHrPrP) containing amino acids 90 to 231 as previously described [11, 15, 22]. In summary, SHrPrP was expressed in BL21 Rossetta *Escherichia coli* (Novagen) cultured in lysogeny broth medium at 37˚C in the presence of selection antibiotics. Bacteria were cultured prior to performing cell lysis with Bugbuster reagent supplemented with Lysonase (EMD Biosciences). Inclusion bodies were collected by centrifugation at 6,000 RPM before solubilizing the pellet overnight (8 M guanidine hydrochloride, 100 mM $Na_2PO_4$). The solubilized rPrP was applied to an XK16-60 column (GE Healthcare) to induce refolding through a linear gradient of denaturation buffer (6 M guanidine hydrochloride, 100 mM $Na_2PO_4$, 10 mM Tris) to refolding buffer (100 mM $Na_2HPO_4$, 10 mM Tris) followed by elution buffer (100 mM NaH2PO4, 10 mM Tris, 0.5 M imidazole) where fractions were collected. The eluted fractions were dialyzed overnight in dialysis buffer (20 mM NaH2PO4). The final concentration of SHrPrP was determined by spectrophotometer, $A_{280,}$ and Beer's Law prior to being stored at 4˚C.

**Assay conditions.** RT-QuIC was performed as previously described [22]. 10% weight/volume tissue homogenates were diluted $10^{-1}$ in 0.1% SDS/1XPBS and 2 uL were added to each well in a 96-well optical bottom plate containing 0.1 mg/ml prion protein substrate and RT-QuIC reaction buffer (20 mM NaH2PO4, 320 mM NaCl, 1.0 mM EDTA, 1 mM thioflavin T). Each RT-QuIC experiment consisted of 250 cycles for a total of 62.5 hours; each cycle included 15 minutes of alternating between 1 minute of shaking at 700 RPM and 1 minute of rest in a BMG Labtech Polarstar$^{TM}$ fluorometer/plate reader. Fluorescence readings were recorded by the reader at the end of each 15 minute cycle with an excitation of 450 nm and emission of 480 nm, using a gain of 1,700.

**RT-QuIC data analysis.** RT-QuIC data from a minimum of two experiments with four replicates each was converted into amyloid formation rates for analysis. Replicates from each sample were considered positive for amyloid formation if the fluorescence rose above the threshold of 5 standard deviations above the average of initial baseline fluorescence readings. The rates of amyloid formation were calculated as previously described [22]. Statistical analysis

of data were performed using GraphPad Prism software. A two-tailed Mann-Whitney test was used to compare samples to corresponding negative controls. P values below 0.05 were considered statistically different. Each RT-QuIC test that is significant compared to controls is marked with ****$p < 0.0001$, ***$p < 0.001$, ** $p < 0.01$, *$p < 0.05$.

## Immunohistochemistry (IHC)

Following paraffin embedding, fixed tissues were processed onto slides by placing 5μm tissue sections onto positively charged slides. Immunodetection of PrP$^{CWD}$ was preformed using a previously described protocol [22]. Following deparaffinization and rehydration with graded alcohols, tissues were treated with 1 μg/ml proteinase K digestion at 37˚C to remove cellular PrP$^C$. Heat-induced epitope antigen retrieval was performed using 10mM EDTA, pH 6.0. Tissues were treated with 88% formic acid for 5 minutes prior to quenching peroxidase activity with 3.0% hydrogen peroxide in methanol. Tissues were blocked with TNB (Perkin-Elmer), then incubated in a primary anti-prion protein antibody BAR224 (Cayman Chemical) at 2 ug/ml overnight. Tissue sections were treated with a secondary goat anti-mouse antibody conjugated to horseradish peroxidase (Abcam). AEC substrate-chromogen (Abcam) was used to detect immunoreactivity and tissue sections were counterstained with Meyer's hematoxylin (Dako) and 0.1% bicarbonate bluing reagent. Slides were cover-slipped using aqueous mounting medium.

## Brain and lymph node tissues

Described tissue preparation as above. CWD infection status of every animal in the study was determined by IHC analysis of fixed tissue sections and RT-QuIC assay of 10% w/v homogenates of brain (obex) and retropharyngeal lymph nodes. IHC and RT-QuIC assay conditions were as described above.

## Acknowledgments

We are thankful for the contributions of Sallie Dahmes at WASCO Inc, and David Osborn, Carl Miller, and Gino D'Angelo at the Warnell School of Forestry and Natural Resources, University of Georgia who provided the deer used in this study and without whom this study would not have been possible.

## Author Contributions

**Conceptualization:** Sarah K. Cooper, Clare E. Hoover, Davin M. Henderson.

**Data curation:** Sarah K. Cooper, Clare E. Hoover, Davin M. Henderson.

**Formal analysis:** Sarah K. Cooper, Clare E. Hoover, Davin M. Henderson.

**Funding acquisition:** Candace K. Mathiason, Edward A. Hoover.

**Investigation:** Sarah K. Cooper, Clare E. Hoover.

**Methodology:** Clare E. Hoover, Davin M. Henderson, Nicholas J. Haley.

**Project administration:** Clare E. Hoover.

**Resources:** Nicholas J. Haley, Candace K. Mathiason, Edward A. Hoover.

**Supervision:** Clare E. Hoover, Davin M. Henderson, Edward A. Hoover.

**Visualization:** Edward A. Hoover.

**Writing – original draft:** Sarah K. Cooper, Clare E. Hoover, Davin M. Henderson.

**Writing – review & editing:** Sarah K. Cooper, Clare E. Hoover, Davin M. Henderson, Edward A. Hoover.

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
