## [Decision Letter · Decision Letter 0]

10 Jul 2019

PONE-D-19-16699

Detection of CWD in Cervids by RT-QuIC Assay of Third Eyelids

PLOS ONE

Dear Dr. Henderson,

Thank you for submitting your manuscript to PLoS One. Your manuscript has been reviewed by two experts in the field, each of whom saw value in your study, as do I. At the same time the reviewers have raised a number of questions and suggestions for improving your manuscript. With respect to Reviewer 2’s comment on Figure 5 and associated text, I have followed up to find out that the problem is with the Reviewer’s difficulty in reconciling all of the numbers in the text with the data in the figure. Accordingly, I encourage you to submit a revised manuscript that addresses the points raised by the reviewers to the best of your ability. If you choose not to follow the suggestions, please explain. 

We would appreciate receiving your revised manuscript by Aug 24 2019 11:59PM. To enhance the reproducibility of your results, we recommend that if applicable you deposit your laboratory protocols in protocols.io, where a protocol can be assigned its own identifier (DOI) such that it can be cited independently in the future. For instructions see: http://journals.plos.org/plosone/s/submission-guidelines#loc-laboratory-protocols

We look forward to receiving your revised manuscript.

Kind regards,

Byron Caughey

Academic Editor

PLOS ONE

Journal Requirements:

Grants:

National Institute of Health

https://www.nih.gov/

R01-NS06190 EAH, P01-AI-077774 EAH, R01-NS076894 EAH, and R01AI112956 CKM.

We note that one or more of the authors are employed by a commercial company: AstraZeneca

Additional Editor Comments:

Thank you for submitting your manuscript to PLoS One. Your manuscript has been reviewed by two experts in the field, each of whom saw value in your study. At the same time the reviewers have raised a number of questions and suggestions for improving your manuscript. With respect to Reviewer 2’s comment on Figure 5 and associated text, I have followed up to find out that the problem is with the Reviewer’s difficulty in reconciling all of the numbers in the text with the data in the figure. Accordingly, I encourage you to submit a revised manuscript that addresses the points raised by the reviewers to the best of your ability. If you choose not to follow the suggestions, please explain.

Reviewers' comments:

Reviewer's Responses to Questions

**Comments to the Author**

1. Is the manuscript technically sound, and do the data support the conclusions?

Reviewer #1: Yes

Reviewer #2: Partly

2. Has the statistical analysis been performed appropriately and rigorously? 

Reviewer #1: Yes

Reviewer #2: No

3. Have the authors made all data underlying the findings in their manuscript fully available?

Reviewer #1: Yes

Reviewer #2: No

4. Is the manuscript presented in an intelligible fashion and written in standard English?

Reviewer #1: Yes

Reviewer #2: No

5. Review Comments to the Author

Reviewer #1: This manuscript applies the third eyelid assay for prions (previously described for scrapie-infected sheep) to the detection of CWD in deer, adding RT-QuIC as a means to enhance detection. The test shows relatively good sensitivity in clinical deer with the 96GG Prnp genotype. The test is less effective in preclinical animals, particularly those with polymorphisms at amino acid 96.

1) A table comparing all the samples would be very helpful (providing times post-infection or symptomatic/pre-symptomatic) to clarify all the various incubation periods/genotypes etc.

2) Were any live animals tested? Will this assay work as an ante-mortem test?

3) Any thoughts as to why the seeding activity is much higher in GS and SS than GG third eyelids late in the disease course but not detectable in the early samples? Does this affect the utility of the test on field samples?

Minor points:

Line 141-142: The sentence "However, RT-QuIC analysis..." needs rewriting, it is awkwardly phrased.

Line 252: Clarification needed--was the test performed by the USDA or just required to be done by the USDA?

Reviewer #2: This study is proposed to diagnose CWD in deer by detecting accumulation of disease associated PrPCWD in the 3rd eyelid lymphoid follicles by RT-QuIC to ease the sample collection protocol over the current protocol which requires detailed anatomical knowledge and dissection skills to collect precise retrophyngeal lymph node tissues. This study showed that PrPCWD detection as early as one month after inoculation of CWD inoculated deer in the 3rd eyelid by RT-QuIC. Interestingly, CWD infected deer with 96GG started accumulating PrPCWD earlier than CWD infected deer with 96GS in the 3rd eyelid. This finding might be expanded in future to address whether the 3rd eyelid is appropriate test samples for all types of CWD infected deer or deer with specific genotypes.

Even though this is an interesting study, insufficient labels and explanations of the figures and table, it is hard to understand the data. For instance, in abstract, line 36 -37 ”IHC detected…5 of 10 deer (50%) whereas third eyelids of all these animals…”, which data shows 5/10 by IHC and 10/10 by RT-QuIC in the third eyelids in clinical deer?

In Introduction, add a rational(s) behind comparing 96GG, GS, and SS genotypes in CWD disease development in deer.

Line 48-50, “ Disease begins when …. (PrPC) into a disease-associated, protease-resistant form (PrPCWD) which….” This is a general definition of PrPRES, and references sited here (12-14) are also covered a general term. So change PrPCWD to PrPRES and add another sentence to define PrPCWD, which is PrPRES found in CWD infected corvid.

Line 64 “…the third eyelid, a nictitating membrane found in….” after defined what is the third eyelid, use one for the rest of the manuscript. For example, Line 65-69 used “the third eyelid” and line 70 used “nictitating membrane” in the same paragraph. Line 120 is another example. Please check the entire manuscript to be consistent in terminology.

Results, request to improve figure and table labels and legends in order to easily understand data.

Figure1A-E

- Explain what was compared using two-tailed Mann-Whitney test?

- Figure 1 B-E, which samples are 96GG or GS? If all GG, explain.

Figure 1B

- 812 looked like a negative control in Fig 1E, are there any reasons behind? Explain.

Line 91 define an abbreviation first time you use, “Brain per os,” oral administration

Line 100-102, hard to follow. Please make points clear.

Line 102 “…in the RPLN, obex, and the third eyelid.” Be consistent in order of the Figure 2, “ the obex, RPLN, and third eyelid…” or change the order of the IHC images.

Line117-119, adding the genotype GG, GS, SS labels to Table 1 will help to understand “4 of 8 96GG deer” data in the table.

Figure 3, report significant values for one star, three stars and four stars.

Line285-289, “A two-tailed Mann-Whitney test was… to corresponding negative controls.” and p-values should be in figure legend.

Explain case 1305 in Figure 4 for lower seeding activity unlike others

Figure 5 and its results section needs to be improved.

The first paragraph in Discussion should be in introduction.

6. PLOS authors have the option to publish the peer review history of their article (what does this mean?). If published, this will include your full peer review and any attached files.

Reviewer #1: No

Reviewer #2: No

---

## [Author Response · Author response to Decision Letter 0]

11 Aug 2019

Dr. Byron Caughey

Re. ms. #PONE-D-19-16699

Dear Dr. Caughey:

We thank the reviewers for their comments and suggestions for revision of this manuscript. We have addressed each comment in the text below and made pertinent changes to the manuscript in accord with their comments.

Our responses are in italics after each reviewer comment.

Reviewer 1:

Reviewer comment 1: A table comparing all the samples would be very helpful (providing times post-infection or symptomatic/pre-symptomatic) to clarify all the various incubation periods/genotypes etc.

Response 1: These points are well taken. We have accordingly replaced Table 1 with Table 2 to include these comparisons to clarify all incubation periods, genotypes, and clinical status.

Reviewer comment 2: Were any live animals tested? Will this assay work as an ante-mortem test?

Response 2: Good question. We did not test live animals as it was important not to risk any negative consequences for longitudinal studies. We suspect the assay would work for live animals using biopsy methods similar to those in O’Rourke et al. 2002 with sheep. For use with RT-QuIC or other amplification assay, new or prion decontaminated instruments and probably short general anesthesia would be required for deer.

Reviewer comment 3: Any thoughts as to why the seeding activity is much higher in GS and SS than GG third eyelids late in the disease course but not detectable in the early samples? Does this affect the utility of the test on field samples?

Response 3: Based on the low number of especially GS animals tested in this study, we are hesitant to conclude the pattern observed is representative overall. The overall similar pattern of systemic lymphoid tissue involvement in 96GG and GS deer also might suggest that this might also be expected for the lymphoid tissue in the nictitating membrane. We also would suspect that less frequent detection of seeding activity in would be expected for early samples from GS or SS deer. 

Minor points: 

Comment: Line 141-142: The sentence "However, RT-QuIC analysis..." needs rewriting, it is awkwardly phrased.

Response: We agree. The wording has been improved to match the syntax of the rest of the paper.

Comment: Line 252: Clarification needed--was the test performed by the USDA or just required to be done by the USDA?

Response: The sentence was restructured to clarify the test was performed and required by the USDA.

Reviewer 2: 

Reviewer 2, comment 1: Even though this is an interesting study, insufficient labels and explanations of the figures and table, it is hard to understand the data. For instance, in abstract, line 36 -37 ”IHC detected…5 of 10 deer (50%) whereas third eyelids of all these animals…”, which data shows 5/10 by IHC and 10/10 by RT-QuIC in the third eyelids in clinical deer?

Response 1: We thank the reviewer for pointing out this lack of clarity. We have accordingly added an additional table to more clearly demonstrate the data described in line 36-37. More specific labels and explanations have been added to all figures and tables to clarify the data presentation. 

Reviewer comment 2: In Introduction, add a rational(s) behind comparing 96GG, GS, and SS genotypes in CWD disease development in deer.

Response 2: In the Introduction we have clarified the importance of genotypes of deer and elk as these polymorphisms pertain to CWD disease development in cervids. (Lines 50 to 54).

Reviewer comment 3: Line 48-50, “ Disease begins when …. (PrPC) into a disease-associated, protease-resistant form (PrPCWD) which….” This is a general definition of PrPRES, and references sited here (12-14) are also covered a general term. So change PrPCWD to PrPRES and add another sentence to define PrPCWD, which is PrPRES found in CWD infected cervid.

Response 3: We appreciate this comment and have adjusted the text referred to in lines 12-14 to properly define PrPCWD.

Reviewer comment 4: Line 64 “…the third eyelid, a nictitating membrane found in….” after defined what is the third eyelid, use one for the rest of the manuscript. For example, Line 65-69 used “the third eyelid” and line 70 used “nictitating membrane” in the same paragraph. Line 120 is another example. Please check the entire manuscript to be consistent in terminology.

Response 4: All uses of nictitating membrane have been replaced with third eyelid to maintain consistent terminology.

Reviewer comment 5: Results, request to improve figure and table labels and legends in order to easily understand data.

Response 5: Thank you. Figure and table labels have been added and improved and legends have been made more specific to improve clarity. 

Reviewer comment 6: Figure1A-E

- Explain what was compared using two-tailed Mann-Whitney test?

- Figure 1 B-E, which samples are 96GG or GS? If all GG, explain.

Response 6: Thank you. We have adjusted each figure legend to clarify what was compared to determine statistical significance. Genotypes have been added to each section of the figure.

Reviewer comment 7: Figure 1B

- 812 looked like a negative control in Fig 1E, are there any reasons behind? Explain.

Response 7: Other tissues of this animal were unequivocally positive for seeding activity and PrPRES. So other than individual animal variation, we do not know the reason why 812 looked like a negative control (i.e. negative) based on third eyelid analysis.

Reviewer comment 8: Line 91 define an abbreviation first time you use, “Brain per os,” oral administration

Response 8: A definition for ‘brain per os’ has been added.

Reviewer comment 9: Line 100-102, hard to follow. Please make points clear.

Response 9: The sentence in question has been re-written to improve clarity. (Lines 116 to 118).

Reviewer comment 10: Line 102 “…in the RPLN, obex, and the third eyelid.” Be consistent in order of the Figure 2, “ the obex, RPLN, and third eyelid…” or change the order of the IHC images.

Response 10: The order of text has been changed to be consistent with the order displayed in Figure 2.

Reviewer comment 11: Line117-119, adding the genotype GG, GS, SS labels to Table 1 will help to understand “4 of 8 96GG deer” data in the table.

Response 11: Thank you for this comment. We have replaced Table 1 with Table 2 to clarify all incubation periods, genotypes, and clinical status.

Reviewer comment 12: Figure 3, report significant values for one star, three stars and four stars.

Response 12: Significant values for 1-4 stars have been reported and added to each figure legend.

Reviewer comment 13: Line285-289, “A two-tailed Mann-Whitney test was… to corresponding negative controls.” and p-values should be in figure legend.

Response 13: A more specific explanation of the statistics performed have been added to each figure legend.

Reviewer comment 14: Explain case 1305 in Figure 4 for lower seeding activity unlike others

Response 14: The lower seeding activity in the third eyelid of animal 1305 we presume to reflect an earlier stage of infection. Unlike other animals in this group, 1305 was positive by IHC only in the retropharyngeal lymph node, whereas other animals in this cohort were IHC-positive in not only in retropharyngeal lymph nodes, but also in other systemic lymph nodes, and the brain.

Reviewer comment 15: Figure 5 and its results section needs to be improved.

Response 15: Figure 5 and its results section (lines 203 to 224) have been revised to improve clarity. 

Reviewer comment 16: The first paragraph in Discussion should be in Introduction.

Response 16: The first paragraph in the Discussion was removed and important aspects were added to the Introduction. (Lines 65 to 70).

Finally, as requested, here we provide additional information in regard to the Funding Statement and Competing Interests Statement. Dr. Clare E. Hoover is currently employed by a commercial company, AstraZeneca. However, this company played no role in this study, did not provide funding for Dr. Clare Hoover’s salary when this work was conducted ,nor did AstraZeneca have any role in the study design, data collection and analysis, decision to publish, nor preparation of the manuscript. The specific roles of this author are articulated in the ‘author contributions’ section.” This does not alter our adherence to PLoS One policies on sharing data and materials.

Thank you for the opportunity to submit our revised manuscript for your further consideration for publication in PLoS One.

Yours sincerely,

Sarah K. Cooper, BS

Graduate Research Associate

Edward A. Hoover, DVM, PhD

University Distinguished Professor

Department of Microbiology, Immunology and Pathology

College of Veterinary Medicine and Biomedical Sciences

---

## [Editor Report · Decision Letter 1]

13 Aug 2019

Detection of CWD in Cervids by RT-QuIC Assay of Third Eyelids

PONE-D-19-16699R1

Dear Dr. Hoover,

We are pleased to inform you that your manuscript has been judged scientifically suitable for publication and will be formally accepted for publication once it complies with all outstanding technical requirements.

With kind regards,

Byron Caughey

Academic Editor

PLOS ONE
---

## [Editor Report · Acceptance letter]

20 Aug 2019

PONE-D-19-16699R1 

Detection of CWD in Cervids by RT-QuIC Assay of Third Eyelids 

Dear Dr. Hoover:

I am pleased to inform you that your manuscript has been deemed suitable for publication in PLOS ONE. Congratulations! Your manuscript is now with our production department. 

With kind regards,

on behalf of

Dr. Byron Caughey 

Academic Editor

PLOS ONE